# Highly Sensitive and Selective Sodium Ion Sensor Based on Silicon Nanowire Dual Gate Field-Effect Transistor

**DOI:** 10.3390/s21124213

**Published:** 2021-06-19

**Authors:** Seong-Kun Cho, Won-Ju Cho

**Affiliations:** Department of Electronic Materials Engineering, Kwangwoon University, 20, Gwangun-ro, Nowon-gu, Seoul 01897, Korea; whtjdrms98@gmail.com

**Keywords:** sodium ion sensor, silicon nanowire, dual-gate field-effect transistor, sodium-selective membrane, extended gate

## Abstract

In this study, a highly sensitive and selective sodium ion sensor consisting of a dual-gate (DG) structured silicon nanowire (SiNW) field-effect transistor (FET) as the transducer and a sodium-selective membrane extended gate (EG) as the sensing unit was developed. The SiNW channel DG FET was fabricated through the dry etching of the silicon-on-insulator substrate by using electrospun polyvinylpyrrolidone nanofibers as a template for the SiNW pattern transfer. The selectivity and sensitivity of sodium to other ions were verified by constructing a sodium ion sensor, wherein the EG was electrically connected to the SiNW channel DG FET with a sodium-selective membrane. An extremely high sensitivity of 1464.66 mV/dec was obtained for a NaCl solution. The low sensitivities of the SiNW channel FET-based sodium ion sensor to CaCl_2_, KCl, and pH buffer solutions demonstrated its excellent selectivity. The reliability and stability of the sodium ion sensor were verified under non-ideal behaviors by analyzing the hysteresis and drift. Therefore, the SiNW channel DG FET-based sodium ion sensor, which comprises a sodium-selective membrane EG, can be applied to accurately detect sodium ions in the analyses of sweat or blood.

## 1. Introduction

Human sweat contains key biomarkers such as glucose, lactate, Na^+^, K^+^, H^+^, NH^4+^, Ca^2+^, and Cl^−^ that correlate with their respective concentrations in blood. Hence, human sweat can be analyzed to conduct real-time, non-invasive personalized health monitoring [1,2]. In particular, it is important to monitor these analytes in patients suffering from complex body conditions, which range from hydration to heart and metabolic diseases. As active electronic devices, field-effect transistor (FET)-based ion sensors are promising candidates for such non-invasive applications owing to their numerous advantages, such as small sizes, portability, and rapid detection [3,4,5,6]. A high-performance ion sensor must possess high sensitivity, stability, and reliability in order to ensure its successful implementation. In addition, the presence of several ions in analytes such as human sweat or blood necessitates high selectivity to ensure that only the target ions are detected. Conventional bulk silicon FET-based sensors with a single-gate (SG) structure exhibit a Nernstian limit of 59.16 mV/dec, which is the theoretical upper limit of sensitivity at room temperature [7,8]. FET-based sensors with dual-gate (DG) structures on silicon-on-insulator (SOI) substrates have also attracted significant attention. In DG-structured ion-sensitive (IS) FETs, capacitive coupling occurs between the top and bottom gate oxide layers of the channel due to structural specificity; this amplifies the sensitivity of these devices [9,10,11,12,13,14]. The amplification ratio increases through capacitive coupling in the DG sensing mode with the increased capacitance of the top gate oxide. We recently reported that a DG-structured silicon nanowire (SiNW) channel FET-based pH sensor exhibits better sensitivity, detection capability, stability, and reliability than conventional FET-based sensors [15]. In previous studies, high sensitivity to pH values was achieved using an SnO_2_ sensing membrane based on the site-binding theory.

In this paper, we propose a highly sensitive sodium ion-selective sensor based on a SiNW channel DG FET. The sensing unit of the extended gate (EG) with a sodium-selective membrane and the transducer unit of the SiNW channel DG FET were built independently. These two units were electrically connected in series to complete the sodium ion sensor platform. The SiNW channel DG FET transducer unit was fabricated through the dry etching of SOI substrates by using electrospun polyvinyl pyrrolidone (PVP) nanofibers (NFs) as a template for the SiNW pattern transfer. The fundamental electrical characteristics and signal amplification functions of the fabricated SiNW channel DG FET were subsequently evaluated. In addition, the sodium-selective membrane EG detects only sodium ions with an excellent selectivity; this is because the sodium ionophore of the membrane can selectively capture sodium ions alone. The sensitivity and ion selectivity of the sodium ion sensor were verified by using NaCl, CaCl_2_, KCl, and pH buffer solutions. Furthermore, the reliability and stability of the sodium ion sensor platform were evaluated by analyzing non-ideal behaviors, such as the hysteresis and drift.

## 2. Materials and Methods

### 2.1. Fabrication of SiNWs on SOI Wafer

The SiNWs were fabricated on a fully depleted p-type (100) SOI wafer with a doping concentration of 1 × 10^15^ cm^−3^, a 750-nm-thick buried oxide (BOX) layer, and a 70-nm-thick top Si layer. The SiNWs were fabricated by transferring the pattern template of the PVP NFs to the uppermost Si layer of the SOI wafer. The PVP NF precursor solution used for electrospinning was prepared by stirring 100 mg of PVP (M_W_ ≈ 1,300,000, Sigma-Aldrich Co., St. Louis, MO, USA) in 1.5 mL of ethanol (≥99.7%) at 800 rpm for 2 h at room temperature. A schematic of the electrospinning system is shown in Figure 1a. It consists of a syringe with a spinning needle, a syringe pump (NE-1000; New Era Pump Systems Inc., Farmingdale, NY, USA), a high-voltage power supply, and a grounded collector (15 × 15 cm^2^ copper plate). The PVP NFs were electrospun onto the SOI substrate by loading a syringe with the PVP NF precursor solution and fixing it horizontally to the syringe pump. The pressure of the syringe pump was 0.4 mL/h; the inner diameter of the needle was 0.635 mm; the working distance between the needle and the grounded collector was 20 cm; the positive voltage was adjusted to 20 kV; and the relative temperature and humidity were maintained at 25 °C and 25%, respectively. Calcination was performed by using a resistive heating furnace at 300 °C for 1 h in air to remove the solvent in the electrospun PVP NFs. The equilibrium-melting temperature of PVP is 300 °C; thus, the abovementioned process improves the thermomechanical stability of the template fibers and the adhesion between the PVP NFs and the substrate surface [16]. An optical microscopy image of the electrospun PVP NFs, which were used as hard masks for the silicon dry etching, is presented in the inset of Figure 1a. A section of the uppermost Si layer was selectively etched by exposing the electrospun SOI substrate with the PVP NFs to the SF_6_ plasma of a reactive ion etching (RIE) system. The power of the plasma treatment in the RIE process was 50 W; the SF_6_ gas flow and pressure were fixed at 20 sccm and 300 mTorr, respectively; and the process duration was 15 s. The residual PVP NFs were removed by using a sulfuric acid–hydrogen peroxide mixture solution after the SF_6_ plasma etching process. A schematic of the fabrication of the SiNWs on an SOI substrate is shown in Figure 1b. Scanning electron microscopy (SEM) images of the patterned SiNWs are presented in Figure 2.

### 2.2. Fabrication of SiNW Channel DG Field-Effect Transistors

The SiNW channel DG FET transducer unit was fabricated by processing the SOI wafer into the SiNW through the transfer of the pattern template of the PVP NFs. The active regions of the FET, which had a channel width and length of 20 and 10 μm, respectively, were formed through photolithography and RIE etching after standard RCA cleaning. Highly doped source/drain regions were selectively formed through the solid-phase diffusion of phosphorus from phosphosilicate glass films with rapid thermal annealing at 950 °C. A 20-nm-thick SiO_2_ layer was deposited as the top gate oxide layer through RF magnetron sputtering. This was followed by the formation of the top gate and the source/drain electrodes, which were 150-nm-thick Al films, through a lift-off process employing an electron beam evaporator. The p-type substrate under the BOX layer of the SOI serves as the bottom gate. Forming gas annealing treatment was performed at 450 °C for 30 min under a 2% H_2_/N_2_ ambient in order to improve the electrical properties and stability of the fabricated SiNW channel DG FET transducer. The schematic of the SiNW channel DG FET transducer unit is presented in Figure 3a.

### 2.3. Fabrication of Sodium-Selective Membrane EG

The sodium-selective membrane EG sensing unit was built by using a 1.5 × 2.5 cm^2^ glass substrate (Corning Inc., New York, NY, USA) to reduce the overall cost of the experiment. A 300-nm-thick ITO conductive film and a 50-nm-thick SnO_2_ film were sequentially deposited through RF magnetron sputtering after cleaning the glass substrate ultrasonically. The sodium-selective membrane was formed by drop-casting 100 μL of the membrane cocktails onto the SnO_2_ film and drying it for 24 h. The sodium-selective membrane cocktail was prepared by stirring 100 mg of a mixture of Na ion channel stage X (1% *w*/*w*), sodium tetrakis [3,5bis(trifluoromethyl)phenyl] borate (Na-TFPB) (0.55% *w*/*w*), polyvinyl chloride (PVC) (33% *w*/*w*), and bis(2-ethylhexyl) sebacate (DOS) (65.45% *w*/*w*) in 660 μL of tetrahydrofuran (THF) at 800 rpm for 4 h at room temperature. In the stacked structure of the ITO/SnO_2_/sodium-selective membrane, the surface potential change of the SnO_2_ film due to the ion capture of the sodium-selective membrane is transferred directly to the uppermost Al gate electrode of the FET through the ITO conductive film. An EG sensing unit was developed by attaching a polydimethylsiloxane (PDMS) chamber with an inner diameter of 0.6 cm to the top of the sodium-selective membrane for solution storage. The reagents were purchased from Sigma-Aldrich Co. (St. Louis, MO, USA) and used without additional purification. A schematic of the sodium-selective membrane EG sensing unit is shown in Figure 3b, and the molecular structure of the Na ionophore X in the sodium-selective membrane is depicted in the inset of Figure 3b.

### 2.4. Characterization of SiNW Channel DG FET and Sodium Ion-Selective Sensor

The current–voltage (I–V) characteristics of the fabricated SiNW channel DG FETs and the sensing properties of the sodium ion sensor, which was constructed by using the SiNW channel DG FET and the sodium-selective membrane EG, were measured using an Agilent 4156B precision semiconductor parameter analyzer. The electrical measurements of these devices were carried out in an electromagnetically shielded dark box to reduce external disturbances, such as noise signals and light. A commercial Ag/AgCl reference electrode (2086A-06T, Horiba, Kyoto, Japan), which was composed of a ceramic-plug junction and an internal solution saturated with KCl and AgCl, was used as the reference electrode to detect ions in the solution. The sensing electrolyte was obtained by dissolving NaCl, KCl, and CaCl_2_ (Sigma-Aldrich Co., St. Louis, MO, USA) in DI water to prepare varying concentrations of NaCl, KCl, and CaCl_2_ solutions, in addition to a commercial pH buffer solution (SAMCHUN CHEMICAL, Pyeongtaek, Korea) containing potassium hydrogen phthalate. Figure 4 shows the electrical connection between the SiNW channel DG FET and the sodium-selective membrane EG for the sodium ion-selective sensor platform. The FET transducer unit has independent gates at the top and bottom of the SiNW channel, thereby enabling the sensing operation through the top or bottom gates. The operation of the top gate, as shown in Figure 4a, was termed as the SG sensing mode, whereas the operation of the bottom gate, as shown in Figure 4b, was termed as the DG sensing mode. The SG sensing mode involved a voltage sweep of the reference electrode connected to the top gate, while the bottom gate was grounded. The DG sensing mode involved a voltage sweep of the bottom gate, while the reference electrode connected to the top gate was grounded. The sensitivity of the sodium ion sensor was determined by varying the voltage in the top or bottom gates (reference voltage, V_ref_) at a drain current of 1 nA (reference current, I_ref_) depending on the concentration of the NaCl solution. In addition, the hysteresis and drift effects were measured to evaluate the non-ideal behaviors of the sodium ion sensor. The hysteresis voltage was equal to the difference between the V_ref_ of the first and last 10^−4^ M in the concentration loop of 10^−4^ → 10^−3^ → 10^−2^ → 10^−1^ → 10^0^ → 10^−1^ → 10^−2^ → 10^−3^ → 10^−4^ M. The drift rate was equal to the magnitude of change in V_ref_ (ΔV_ref_) due to the exposure of the sodium-selective membrane to a 10^−4^ M NaCl solution for 10 h.

## 3. Results and Discussion

### 3.1. Electrical Characteristics of SiNW Channel DG FETs

Figure 5 shows the transfer characteristic (I_D_–V_G_) curves and output characteristic (I_D_–V_D_) curves (insets) of the SiNW channel DG FETs, where Figure 5a,b correspond to the top gate and bottom gate operation modes, respectively. The transfer characteristics were measured at drain voltages (V_D_) of 50 mV and 1 V for the top gate and bottom gate operation modes, respectively. The top gate voltage was varied from −1.0 to 1.5 V (in steps of 25 mV) to measure the transfer characteristics of the top gate operation. Similarly, the bottom gate voltage was varied from −15 to 35 V (in steps of 500 mV) for the bottom gate operation. The output characteristics were obtained by varying the top gate voltage from 0 to 1 V (in steps of 0.1 V) and the bottom gate voltage from 0 to 30 V (in steps of 3 V). As a result, the SiNW channel FETs, which were based on the electrospun PVP NF pattern template transfer method, exhibited excellent electrical characteristics during the top and bottom gate operations.

Table 1 lists the electrical parameters extracted from the transfer characteristic curves of the SiNW channel DG FETs. A threshold voltage (V_TH_) of −0.3 V, field-effect mobility (μ_FE_) of 713.1 cm^2^/V·s, sub-threshold swing (SS) of 96.9 mV/dec, and an on/off current ratio (I_ON_/I_OFF_) of 3.0 × 10^7^ were obtained during the top gate operation. Similarly, the values of V_TH_, µ_FE_, SS, and I_ON_/I_OFF_ were −2.0 V, 956.1 cm^2^/V·s, 713.8 mV/dec, and 6.4 × 10^7^, respectively, under the bottom gate operation. These values confirm the excellent electrical performance of the SiNW channel DG FET, which is the basic component of high-sensitivity SG or DG mode sensing and selective sodium sensors.

Additionally, to investigate the effect on noise, we tested the stability of the two modes, as shown in Figure 6, by applying a 0.4 V amplitude square wave pulse (assuming noise) to the gate voltage sweep. As a result, I_max_/I_min_ = 78.36 in the SG mode, while I_max_/I_min_ = 1.3 in the DG mode. This is because the S.S of the transfer characteristic curve in SiNW channel DG mode is 713.8 mV/dec, which is larger than the SS of 96.9 mV/dec in SG mode, and the fluctuation of the drain current with respect to the pulse (noise) of the gate voltage is relatively small. Therefore, we verified that SiNW channel DG FET have better noise tolerance than conventional SG FETs.

### 3.2. Signal Amplification Capabilities of SiNW Channel DG FETs

The maximum sensitivity of an SG FET-based sensor is limited to 59.16 mV/dec at room temperature, because the sensitivity ($\Delta V_{TH}^{top}$) is determined according to the surface potential ($\Delta \psi_{0}$) of the sensing membrane: $\Delta V_{TH}^{top}=-\Delta \psi_{0}$ [17,18]. This value is known as the Nernstian limit. The sensitivity of a DG FET-based sensor can also be amplified through capacitive coupling between the top and bottom gate oxides of the channel due to their structural specificity [17,18,19].
(1)$$\Delta V_{TH}^{bottom}=-\frac{C_{top}}{C_{bottom}}\Delta \psi_{0}=\frac{C_{top}}{C_{bottom}}\Delta V_{TH}^{top},$$
where $C_{top}$ and $C_{bottom}$ represent the capacitance per unit area of the top and bottom gates, respectively. $\Delta V_{TH}^{bottom}$ and $\Delta V_{TH}^{top}$ represent the change in the *V_TH_* of the bottom and top gates, respectively; these are indicators of sensitivity. The abovementioned relationship suggests that sensitivity can significantly exceed the Nernstian limit if an appropriate value of $\Delta V_{TH}^{bottom}$ and a suitable combination of $C_{top}$ and $C_{bottom}$ are employed. The capacitive coupling ratio, which is essentially the ratio of the amplification, increases if the magnitude of $C_{top}$ is greater than that of $C_{bottom}$. As can be seen from Equation (1), reducing the thickness of the top gate oxide increases $C_{top}$, which improves the capacitive coupling. However, as the thickness of the top gate oxide decreases, reliability issues arise owing to the increased leakage current or the poor dielectric breakdown properties of the thin gate insulator. Meanwhile, the SiNW-type channel has a larger specific surface area than the conventional planar film-type channel, resulting in a larger $C_{top}$. Therefore, the SiNW-type channel is an effective structure that can improve the capacitive coupling without reducing the thickness of the top gate oxide. Thus, SiNW-type channels feature larger capacitive coupling ratios than film-type channels; this has also been confirmed in a previous study [15]. The transfer characteristics during the bottom gate operation were measured by using a constant top gate bias to evaluate the signal amplification capabilities of the SiNW channel DG FETs, as shown in Figure 7a. The top gate bias was varied from +600 to −600 mV in steps of 300 mV. A reduction in the top gate bias gradually shifts the transfer characteristic curve to a relatively large bottom gate voltage. Figure 7b shows the variation in the top gate bias with the ΔV_ref_ of the SiNW channel DG FETs, which was obtained from the transfer characteristic curves. The ratio of the change in ΔV_ref_ to the top gate bias, which represents the capacitive coupling ratio of the SiNW channel DG FETs, was 26.3. This implies that the SiNW channel DG FET can amplify the conventional sensitivity by a factor of 26.3, thereby improving the sensitivity in the DG sensing mode significantly.

### 3.3. Signal Amplification Capabilities of SiNW Channel DG FETs

The Na ionophore X of the sodium-selective membrane captures the sodium ions from NaCl molecules and modifies the $\psi_{0}$ of the SnO_2_-sensing dielectric film, upon the exposure of the sodium-selective membrane of the EG to a solution of NaCl [20,21]. The $\Delta \psi_{0}$ of the SnO_2_-sensing dielectric film is applied to the Al top gate of the FET transducer, which is electrically connected in series through the ITO conductive film of the EG. Therefore, the sensitivity of the SG mode depends only on $\psi_{0}$, whereas the sensitivity of the DG mode is given by the value of $\psi_{0}$ amplified by the capacitive coupling ratio, as shown in Equation (1).

Figure 8 shows the transfer characteristic curves of a sodium ion sensor based on a SiNW channel DG FET with a sodium-selective membrane EG. The sensing characteristics for the NaCl, CaCl_2_, KCl, and pH buffer solutions during the SG mode operations are shown in Figure 8a–d, respectively. Figure 8e–h present the sensing characteristics during the DG mode operations for the same solutions. It is noteworthy that the distinct shift in the transfer characteristic curve of the sensing characteristics was only observed for the NaCl solution. It means that the sodium-selective membrane only reacts with the sodium ions of NaCl. A reduction in the concentration shifts the transfer curves in the positive direction.

Figure 9a,b, which were obtained from the transfer characteristic curves of the SG and DG mode operations, respectively, display the variation in the electrolyte concentration with the ΔV_ref_ of the sodium ion sensor. The slope of linear fitting of the ΔV_ref_ plot represents the sensitivity. As can be seen in Figure 8a,e, the transfer characteristic curves shifted at regular intervals only in the 10^−4^~10^0^ M section. It showed poor linearity at 10^−5^ M, which means that the limit of detection is about 10^−4^ M. Therefore, we extracted the sensitivity in the 10^−4^~10^0^ M section. The SiNW channel FET-based sodium ion sensor implemented in this study has a sensitivity of 57.33 mV/dec with respect to NaCl in the SG mode, which is similar to that of a conventional FET-based sodium sensor. However, an extremely high sensitivity of 1464.66 mV/dec was achieved in the DG mode, which is 26.6 times the maximum sensitivity of a conventional FET-based sodium sensor (~55 mV/dec). As a result of evaluating the Na-sensing ability for mV by taking the reciprocal of the sensitivity, the SG mode and DG mode are 1.74 × 10^−2^ dec/mV and 6.85 × 10^−4^ dec/mV, respectively. The goal of sensitivity amplification is to reduce resolution, which means the smallest change in concentration that the sensor can detect. Therefore, in terms of the resolution of the sodium ion sensor, it is evident that DG mode operation is much more advantageous than SG mode operation due to the capacitive coupling. In addition, the linearity of the ΔV_ref_ plots including 100 mM and 200 mM, which is related to physiological concentration, was 99.7% and 99.8% in SG mode and DG mode, respectively. Linearity means how linearly the ΔV_ref_ plot is distributed and how ΔV_ref_ is constantly increasing. The slope of the ΔV_ref_ plot fitting line represents the sensitivity and means the mV shift of the transfer characteristic curve per concentration. Therefore, ΔV_ref_ for different concentrations can be predicted through sensitivity, and the higher the linearity of the ΔV_ref_ plots, the better the reliability. The sodium ion sensor proposed in this paper showed excellent sensing performance with high sensitivity and excellent linearity. Moreover, the responses of the SiNW channel FET-based sodium ion sensor to the CaCl_2_, KCl, and pH buffer solutions were relatively insignificant. The sensitivities in the SG and DG modes did not exceed the Nernstian limit. Therefore, the proposed sodium ion sensor is highly selective to sodium owing to its extremely high sensitivity to NaCl buffer solutions, especially in the DG mode. However, its sensitivity to the CaCl_2_, KCl, and pH buffer solutions was considerably low, which implies that the sodium-selective membrane has excellent selectivity for other ions.

In addition, based on comprehensive theoretical analyses of DG structures, previous studies have reported that DG FET-based sensors outperform SG FET-based sensors in terms of the signal-to-noise ratio (SNR) and resolution [22]. The SNR of a sensor is expressed as the ratio of signal power to the noise power. Although instrument noise is similar for both SG FET and DG FET, DG FET is more advantageous in terms of SNR because DG FET has higher sensitivity than SG FET. Therefore, SiNW channel DG FETs not only achieve high sensitivity through capacitive coupling, but also serve as a stable sensor platform against noise.

### 3.4. Reliability and Stability Evaluation of Sodium Ion Sensors

As the non-ideal operations of chemical sensors limit their accuracies, it is important to ensure their reliability and stability in repetitive and long-term operations, in order to implement high-performance sensors. In this study, the hysteresis and drift effects were measured to evaluate the reliability and stability of the sodium ion sensor based on the SiNW channel DG FET with a sodium-selective membrane EG. The hysteresis effect was used to evaluate reliability by measuring the variation in the sensing properties of the sensor under exposure to varying solution concentrations. Similarly, the drift effect was used to evaluate stability by measuring the variation in the sensing behavior of the sensor under prolonged exposure [23,24,25,26].

Figure 10a,b show the hysteresis effects of a sodium ion sensor based on a SiNW channel DG FET with a sodium-selective membrane EG operating in the SG and DG modes, respectively. Following a concentration loop of 10^−4^ → 10^−3^ → 10^−2^ → 10^−1^ → 10^0^ → 10^−1^ → 10^−2^ → 10^−3^ → 10^−4^ M, the hysteresis voltages were 2.5 mV and 12.4 mV in the SG and DG modes, respectively. These values were significantly lower than its sensitivity to sodium. The hysteresis voltages of the SG and DG modes were 4.3% and 0.8% of the sensitivity voltage, respectively. This indicates that the SiNW channel DG FET-based sodium ion sensor with a sodium-selective membrane EG is more reliable in the DG mode than in the SG mode owing to its high sensitivity.

Figure 11a,b show the drift effects of the sodium ion sensor based on a SiNW channel DG FET with a sodium-selective membrane EG operating in the SG and DG modes, respectively. The drift rates of the sodium ion sensor after exposure to a sodium electrolyte concentration of 10^−4^ M for 10 h were 2.7 mV/h and 17.2 mV/h in the SG and DG modes, respectively. The drift rates in the SG and DG modes were low in comparison to the sensitivity, similar to the observations for the hysteresis voltage. The drift rates in the SG and DG modes were 4.7% and 1.2% of the sensitivity, respectively. This indicates that the SiNW channel DG FET-based sodium ion sensor with a sodium-selective membrane EG is more stable and reliable in the DG mode than in the SG mode owing to its relatively high sensitivity in the former.

## 4. Conclusions

In this paper, a highly sensitive sodium ion sensor with excellent sensitivity was proposed; the sensor comprised a SiNW channel DG FET and a sodium-selective membrane EG as the transducer and sensing units, respectively. A DG structure FET consisting of a SiNW channel was fabricated as a transducer device by using a simple and inexpensive PVP NF pattern template transfer method, in order to maximize capacitive coupling and amplify sensitivity through capacitive coupling. The fabricated SiNW channel DG FET exhibited excellent electrical performance and signal amplification. In addition, a sodium-selective membrane EG was fabricated with a sensing device that selectively detected only sodium ions; the Na ionophore X in the sodium-selective membrane only captured the sodium ions from solutions and altered the surface potential of the SnO_2_ sensing dielectric films. A sodium ion sensor was constructed by electrically connecting the sodium-selective membrane EG to the SiNW channel DG FET in series. An extremely high sensitivity of 1464.66 mV/dec was obtained for NaCl solutions in the DG mode. However, the sensitivities to the CaCl_2_, KCl, and pH buffer solutions were significantly lower than that for the NaCl buffer solution. This proved the excellent selectivity of the sensor. Furthermore, the hysteresis voltage and the drift rate under non-ideal conditions were 12.4 mV and 17.2 mV/h, respectively, thereby confirming the excellent reliability and stability of the sensor, in addition to its enhanced sodium ion sensitivity. These results suggest that the proposed sodium ion sensor is not only highly sensitive and selective toward sodium ions, but also reliable for repetitive and long-term operations. Thus, the SiNW channel DG FET-based sodium ion sensor platform with the sodium-selective membrane EG can be used for the precise and selective detection of sodium ions in the bioanalysis of sweat or blood.

## Figures and Tables

**Figure 1 sensors-21-04213-f001:**
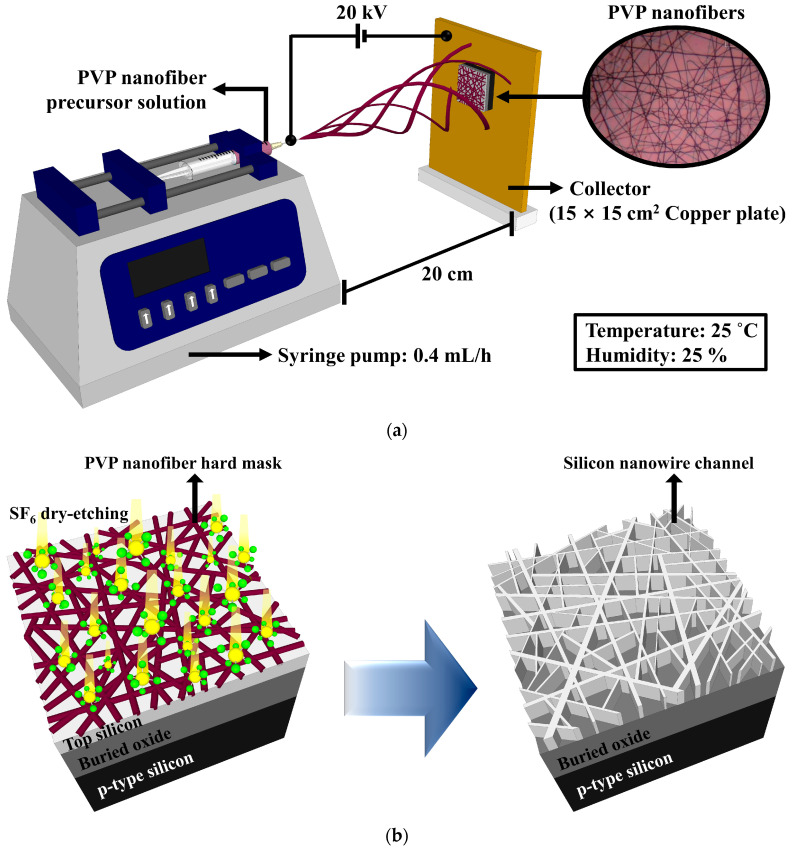
(**a**) Schematic of the electrospinning system. (**b**) Fabrication of SiNWs on an SOI substrate.

**Figure 2 sensors-21-04213-f002:**
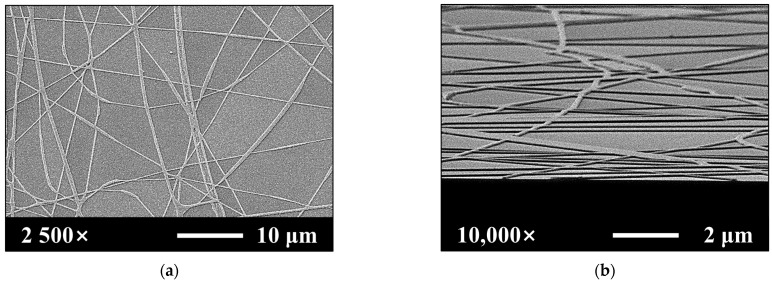
(**a**) Top view and (**b**) tilted view SEM images of the SiNWs after transferring the pattern template of the PVP NFs to the uppermost Si layer of the SOI wafer.

**Figure 3 sensors-21-04213-f003:**
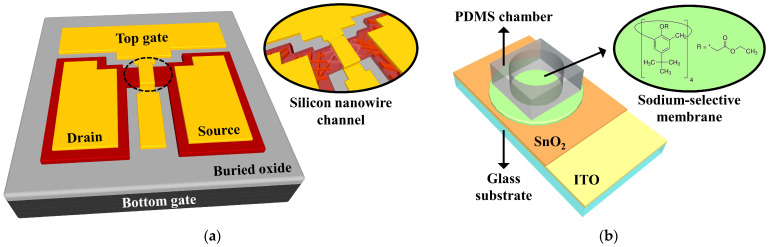
Schematic of the (**a**) SiNW channel DG FET transducer unit and the (**b**) sodium-selective membrane EG sensing unit.

**Figure 4 sensors-21-04213-f004:**
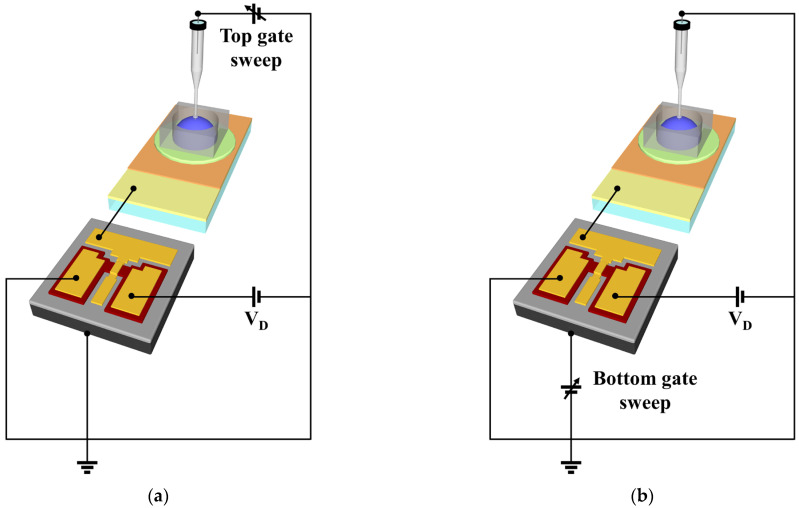
Schematic of the (**a**) SG mode and (**b**) DG mode sensing operations.

**Figure 5 sensors-21-04213-f005:**
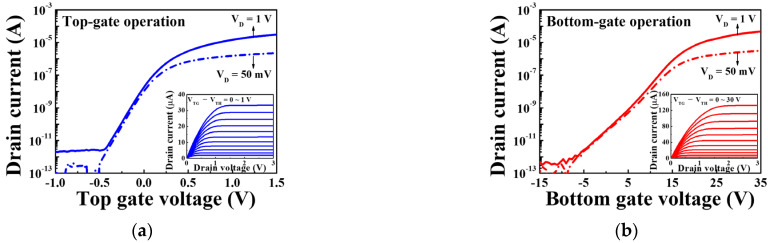
Transfer characteristic curves for the SiNW channel DG FETs operated by the (**a**) top gate and (**b**) bottom gate. The insets display the output characteristic curves.

**Figure 6 sensors-21-04213-f006:**
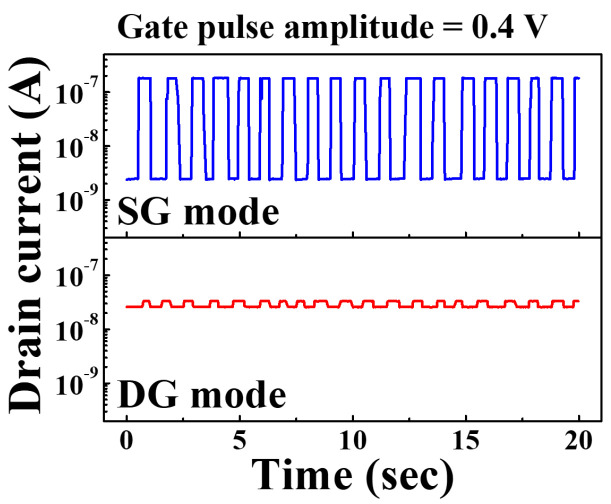
The fluctuation of drain current due to square wave pulse (assuming noise) applied to the voltage sweep under SG and DG operation mode.

**Figure 7 sensors-21-04213-f007:**
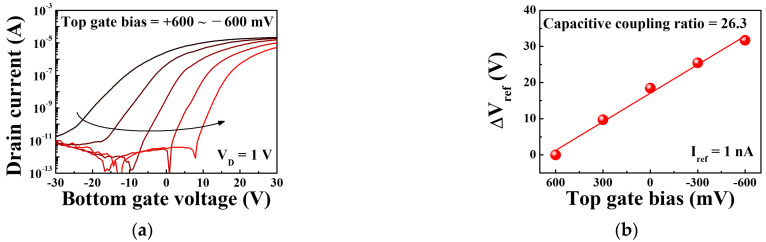
(**a**) Transfer characteristic curves of the SiNW channel DG FET during the bottom gate operation with a constant top gate bias varying from +600 to −600 mV (in steps of 300 mV). (**b**) Variation in the top gate bias with ΔV_ref_.

**Figure 8 sensors-21-04213-f008:**
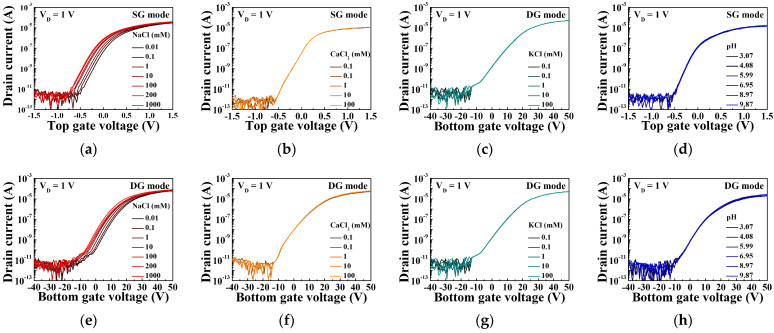
Transfer characteristic curves of the sodium ion sensor based on a SiNW channel DG FET with a sodium-selective membrane EG in various buffer solutions: SG mode operations for the (**a**) NaCl, (**b**) CaCl_2_, (**c**) KCl, and (**d**) pH buffer solutions, and DG mode operations for the (**e**) NaCl, (**f**) CaCl_2_, (**g**) KCl, and (**h**) pH buffer solutions.

**Figure 9 sensors-21-04213-f009:**
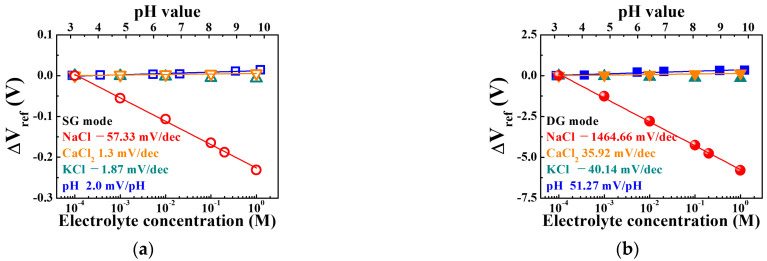
Sensitivity of the sodium ion sensor based on a SiNW channel DG FET with sodium-selective membrane EG in various buffer solutions during the (**a**) SG mode and (**b**) DG mode operations.

**Figure 10 sensors-21-04213-f010:**
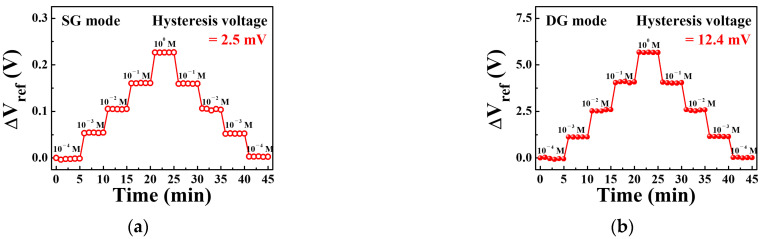
Hysteresis effects of the sodium ion sensor based on SiNW channel DG FETs with sodium-selective membrane EG during the (**a**) SG mode and (**b**) DG mode operations.

**Figure 11 sensors-21-04213-f011:**
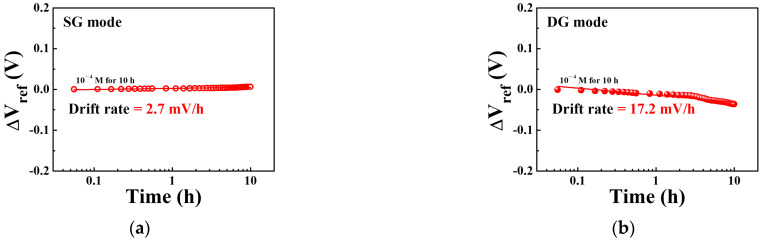
Drift effects of the sodium ion sensor based on SiNW channel DG FETs with a sodium-selective membrane EG after exposure to a sodium electrolyte concentration of 10^−4^ M for 10 h during the (**a**) SG mode and (**b**) DG mode operations.

**Table 1 sensors-21-04213-t001:** Electrical parameters of the SiNW channel DG FETs during top and bottom gate operations.

Operation Mode	V_TH_ (V)	μ_FE_ (cm^2^/V·s)	SS (mV/dec)	I_ON_/I_OFF_ (A/A)
Top gate	−0.3	713.1	96.9	3.0 × 10^7^
Bottom gate	−2.0	956.1	713.8	6.4 × 10^7^

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
