# Peer review of "Highly Sensitive and Selective Sodium Ion Sensor Based on Silicon Nanowire Dual Gate Field-Effect Transistor"

_sensors, 2021, doi:10.3390/s21124213_

Round 1

Reviewer 1 Report

The authors Seong-Kun Choet al in here present a highly sensitive and selective sodium ion sensor consisting of a dual gate (DG) structured silicon nanowire (SiNW) field-effect transistor (FET) as the transducer and a sodium-selective 9 membrane extended gate (EG) as the sensing unit. Plus, an extremely high sensitivity of 1460.08 mV/dec was obtained for the NaCl solution. This is a very promising method, the SiNW channel DG FET-based sodium ion sensor, with a sodium-selective membrane EG, can be  used to detect sodium ions accurately in the analysis of sweat or blood. In view of developing Sensor Based on Silicon Nanowire Dual Gate Field-Effect Transistor is a topic of an incessant research interest, I recommend to publish after very careful revision.

  1. Compared to recently published papers such as Traiwatcharanon, Pranlekha, Wilai Siriwatcharapiboon, and Chatchawal Wongchoosuk. "Electrochemical Sodium Ion Sensor Based on Silver Nanoparticles/Graphene Oxide Nanocomposite for Food Application." Chemosensors8.3 (2020): 58.; Cho, Seong-Kun, and Won-Ju Cho. "Ultra-high sensitivity pH-sensors using silicon nanowire channel dual-gate field-effect transistors fabricated by electrospun polyvinylpyrrolidone nanofibers pattern template transfer." Sensors and Actuators B: Chemical 326 (2021): 128835.;, the principle and innovation of this paper should be explained more in details.
  2. in page 3 figure 2a, the color of words is recommended to be revised
  3. There are many abbreviations in use. Please show the full names before use them.
  4. There are many grammar errors which should be revised carefully.

Author Response

Response Letter to Reviewer #1
We sincerely thank you for giving us such valuable revision suggestions. The reviewers′ comments are really constructive and helpful for us to improve the manuscript. Therefore, we correctly revised the manuscript based on the reviewers′ comments. The following is our point-to-point response to the reviewers′ concerns and descriptions of the revisions, highlighted in yellow in the revised manuscript: 

[Comment 1]
Compared to recently published papers such as Traiwatcharanon, Pranlekha, Wilai Siriwatcharapiboon, and Chatchawal Wongchoosuk. "Electrochemical Sodium Ion Sensor Based on Silver Nanoparticles/Graphene Oxide Nanocomposite for Food Application." Chemosensors 8.3 (2020): 58.; Cho, Seong-Kun, and Won-Ju Cho. "Ultra-high sensitivity pH-sensors using silicon nanowire channel dual-gate field-effect transistors fabricated by electrospun polyvinylpyrrolidone nanofibers pattern template transfer." Sensors and Actuators B: Chemical 326 (2021): 128835.;, the principle and innovation of this paper should be explained more in details.

[Answer 1]
Thank you for your important advice on our manuscript. A reviewer-mentioned first paper by Traiwatcharanon et al. in Chemosensors 8.3 (2020): 58. is screen-printed silver electrode-based sodium ion sensors using silver nanoparticles/graphene oxide nanocomposites, which is a passive electronic device with a resistor-type sensor platform that detects changes in electrode resistance with electrolyte concentration. In comparison, the sodium ion sensor proposed in this study is a silicon-based FET-type sensor platform corresponding to an active electronic device. In addition, the sodium ion sensor proposed in this study has the advantage of an active device that greatly amplifies the sensed signal, and high sensitivity is achieved due to its structural specificity. Meanwhile, the second paper referred by the reviewer, Sensors and Actuators B: Chemical 326 (2021), 128835. by Cho et al., used a silicon nanowire channel double gate FET as a transducer unit and a SnO2 sensing membrane-based extended gate as a sensing unit, demonstrating the effect of nanowire channels and pH sensing. In this paper, as a more advanced study, we implemented “sodium ion sensors” with excellent sodium ion selectivity and high sensitivity by using a “sodium-selective membrane” in a silicon nanowire channel double gate FET transducer. Eventually, the sodium ion sensor proposed in this study selectively detects only sodium ions through a sodium-selective membrane, and greatly amplifies the detected signal through a silicon nanowire channel dual gate FET to achieve high sensitivity.

In response to the reviewer’s comments, we have also revised the manuscript in the following section for the reader's understanding:

(Line 30 – Line 32, Page 1 in revised manuscript)
As active electronic devices, field-effect transistor (FET)-based ion sensors are promising candidates for such non-invasive applications owing to their numerous advantages, such as small sizes, portability, and rapid detection [3–6].

(Line 46 – Line 47, Page 2 in revised manuscript)
In previous studies, high sensitivity to pH values was achieved using an SnO2 sensing membrane based on the site-binding theory. 

(Line 56 – Line 58, Page 2 in revised manuscript)
In addition, the sodium-selective membrane EG detects only sodium ions with an excel-lent selectivity; this is because the sodium ionophore of the membrane can selectively capture sodium ions alone. 

[Comment 2]
In page 3 figure 2a, the color of words is recommended to be revised.

[Answer 2]
Thank you for your valuable comment. Following the reviewer’s comment, we corrected the words in the figures to be clearer. 

[Comment 3]
  There are many abbreviations in use. Please show the full names before use them.

[Answer 3]
Thank you for your helpful comment. We reviewed and carefully revised the manuscript.

[Comment 4]
There are many grammar errors which should be revised carefully.

[Answer 4]
Thank you for your helpful comment. We reviewed the manuscript and carefully revised it.

Again, thank you for your kind consideration and significant advice to our manuscript.

Sincerely yours,
Won-Ju Cho
Department of Electronic Materials Engineering, Kwangwoon University,
20, Gwangun-ro, Nowon-gu, Seoul, 01897, Republic of Korea
E-mail: [email protected]

Reviewer 2 Report

Summary:

The principal aim of the work is the implementation and characterisation of a highly sensitive and selective sodium ion sensor. The sensor consists of a transducer (FET) and a sensing (Na permeable membrane) unit. A dual gate FET consisting of Si nanowires and Al electrodes has been realised and characterised. The Si nanowires have been fabricated by RIE a SOI substrate using an electro-spun PVP template and subsequent deposition of the Al electrodes. The sensing unit consists of a ITO/SnO2 electrode covered by a sodium selective membrane. Compared to the single gate, through capacitive coupling, the sensitivity of the dual gate FET is substantially enhanced. Contrasting and long-term experiments revealed high selectivity for Na and good stability, respectively, of the sensor. In addition, cyclic measurements confirmed low hysteresis values.

Comments:

Alkali metals play a crucial role for many physiological processes. Therefore, selective measurement of sodium ions is of high importance. An interesting approach is presented in this article, however, fabrication and characterisation of the dual FET configuration has already been published in a previous article. The novelty of the work is the selective measurement of sodium ions. The work has been soundly carried out but some important parameters, such as LOD, resolution repeatability, etc. are missing. A high sensitivity has been achieved but a more relevant parameter is the signal-to-noise ratio. As the authors target biological applications, physiological relevant experiments should be carried out or at least discussed. For instance, using this sensor is the resolution good enough to measure physiological relevant changes in the sodium concentration of blood or sweat?

 In addition, only one single configuration has been investigated. Could the performance be further improved changing, for example, the thickness of the oxide layers? These aspects should at least be discussed.

The content of the article is interesting and has potential for applications but, as the dual FET part has already been published, the substance is rather deficient.

General remarks:

  • The manuscript is well structured and clearly written
  • The figures are explicit and the captures are adequate
  • In figure 8, 9 and 10 the experimental conditions could eventually be added

Author Response

Response Letter to Reviewer #2
We sincerely thank you for giving us such valuable suggestions for revision. The reviewers′ comments are indeed constructive and helpful for us to improve our manuscript. Therefore, we correctly revised the manuscript based on the reviewers′ comments. The following is our point-to-point response to the reviewers′ concerns and descriptions of the revisions, highlighted in yellow in the revised manuscript: 

[Comment 1]
Alkali metals play a crucial role for many physiological processes. Therefore, selective measurement of sodium ions is of high importance. An interesting approach is presented in this article, however, fabrication and characterisation of the dual FET configuration has already been published in a previous article. The novelty of the work is the selective measurement of sodium ions. The work has been soundly carried out but some important parameters, such as LOD, resolution repeatability, etc. are missing. A high sensitivity has been achieved but a more relevant parameter is the signal-to-noise ratio. As the authors target biological applications, physiological relevant experiments should be carried out or at least discussed. For instance, using this sensor is the resolution good enough to measure physiological relevant changes in the sodium concentration of blood or sweat?

[Answer 1]
We admired the reviewer's sharp and helpful questions. Resolution is a very important factor in the sensor, and for this, the signal-to-noise ratio must be evaluated. To evaluate this, the sensor resolution under single gate (SG) and dual gate (DG) operation during the same time period can be explained as follows. As shown in Figure 8 of manuscript, SG mode has ΔVref (change in response voltage) of 57.15 mV per decade, while DG mode has ΔVref of 1460.08 mV per decade. Reversely extracting the Na-sensing ability for mV results in SG = 1.74 × 10-2 dec/mV and DG = 6.85 × 10-4 dec/mV. These results were detected during the same measurement time. The ultimate goal of amplifying sensitivity is to improve the corresponding resolution, that is, to reduce the minimum concentration change that the sensor can detect. Therefore, it is evident that the DG mode operation in terms of the sensitivity resolution of the sodium ion sensor is significantly greater than the SG mode operation, which is attributed to capacitive coupling.
Additionally, to investigate the effect on noise, we tested the stability of the two modes as shown in Figure R1 by applying a 0.4 V amplitude square wave pulse (assuming noise) to the gate voltage sweep. As a result, in the case of the SiNW channel DG FET, Imax/Imin = 78.36 in the SG mode, while Imax/Imin = 1.3 in the DG mode. This is because the subthreshold swing (S.S) of the transfer characteristic curve in SiNW channel DG mode is 713.8 mV/dec, which is larger than the SS of 96.9 mV/dec in SG mode, and the fluctuation of the drain current with respect to the pulse (noise) of the gate voltage is relatively small. Therefore, we verified that SiNW channel DG FET-based sodium ion sensors have better noise tolerance than conventional SG FETs. Meanwhile, as a result of comprehensive theoretical analysis of the dual-gate sensor in Reference 1, it was reported that the DG FET-based sensors have better signal-to-noise ratio and smaller resolution than the SG FET-based sensors. [R1]. The following Figure R2 and Table R1 summarize the results of a comprehensive theoretical analysis of the double gate sensor described in Reference 1.

Figure R1. The fluctuation of drain current due to square wave pulse (assuming noise) applied to the voltage sweep under SG and DG operation mode.

Figure R2. Illustrations of signal (S, sensitivity shown as gray bars) and noise (δV shown as yellow bars) of ISFET/DGFET pH sensors: (a) negligible instrument noise (δVIns) and (b), (c) with instrument noise larger than the ISFET noise (δVIns). [R1] 
Table R1. Summary of SNR for ISFET and DGFET sensors in three cases [R1].

In response to the reviewer’s comments, we have also revised the manuscript in the following section for the reader's understanding:

(Line 258 – Line 265, Page 8 in revised manuscript)
In addition, based on comprehensive theoretical analyses of DG structures, previous studies have reported that DG FET-based sensors outperform SG FET-based sensors in terms of the signal-to-noise ratio (SNR) and resolution [22]. The SNR of a sensor is ex-pressed as the ratio of its sensitivity to noise. When the instrument noise is greater than the intrinsic noise of an SG FET or DG FET, the DG FET is superior to the SG FET in terms of the SNR. This is because, although the instrument noise is the same for both the devices, the DG FET features higher sensitivity. Therefore, the SiNW channel DG FET achieves high sensitivity, serving as a stable sensor platform against noise.

[Comment 2]
In addition, only one single configuration has been investigated. Could the performance be further improved changing, for example, the thickness of the oxide layers? These aspects should at least be discussed.

[Answer 2]
We fully agree with your comment. In Equation (1) of the manuscript, it can be seen that capacitive coupling is improved with the increase of Ctop. However, while reducing the thickness of the top gate oxide can improve capacitive coupling, there are reliability problems due to increased leakage current or dielectric breakdown in thin gate insulators. Meanwhile, the capacitive coupling can be improved by applying a high-k material to the top gate oxide. However, in this case, there are problems such as threshold voltage control, carrier mobility degradation, charge trapping, gate insulator breakdown, and temperature instability. Therefore, in this study, the capacitive coupling was improved by introducing a SiNW channel to overcome these problems and limitations. Since the SiNW-type channel has a large specific surface area, it is an efficient structure that can enhance capacitive coupling by increasing Ctop. 

In response to the reviewer’s comments, we have also revised the manuscript in the following section for the reader's understanding:

(Line 201 – Line 208, Page 7 in revised manuscript)
As can be seen from Equation (1), reducing the thickness of the top gate oxide increases Ctop, which improves the capacitive coupling. However, as the thickness of the top gate oxide decreases, reliability issues arise owing to the increased leakage current or the poor dielectric breakdown properties of the thin gate insulator. Meanwhile, the SiNW-type channel has a larger specific surface area than the conventional planar film-type channel, resulting in a larger Ctop. Therefore, the SiNW-type channel is an effective structure that can improve the capacitive coupling without reducing the thickness of the top gate oxide.

[Reference]
[R1]    Go, J.; Nair, P. R.; Alam, M. A. Theory of signal and noise in double-gated nanoscale electronic pH sensors. J. Appl. Phys. 2012, 112(3), 034516.

Again, thank you for your kind consideration and significant advices of our manuscript.

Sincerely yours,
Won-Ju Cho
Department of Electronic Materials Engineering, Kwangwoon University,
20, Gwangun-ro, Nowon-gu, Seoul, 01897, Republic of Korea
E-mail: [email protected]

Round 2

Reviewer 2 Report

Titel: Highly Sensitive and Selective Sodium Ion Sensor Based on Silicon Nanowire Dual Gate Field-Effect Transistor

Summary:

The principal aim of the work is the implementation and characterisation of a highly sensitive and selective sodium ion sensor. The sensor consists of a transducer (FET) and a sensing (Na permeable membrane) unit. A dual gate FET consisting of Si nanowires and Al electrodes has been realised and characterised. The Si nanowires have been fabricated by RIE a SOI substrate using an electro-spun PVP template and subsequent deposition of the Al electrodes. The sensing unit consists of a ITO/SnO2 electrode covered by a sodium selective membrane. Compared to the single gate, through capacitive coupling, the sensitivity of the dual gate FET is substantially enhanced. Contrasting and long-term experiments revealed high selectivity for Na and good stability, respectively, of the sensor. In addition, cyclic measurements confirmed low hysteresis values.

Comments:

The authors introduced a few improvements to the manuscript. They present an interesting approach for sodium ion sensing and propose applications in the field of biosensing. The dual gate configuration has already been published. The novelty of the article is the introduction of the sodium selective membrane which is sparsely discussed. Especially a physiological context is completely missing. What are physiological relevant concentrations, is the sensor stable enough for the detection of physiological relevant changes, LOD, noise and the signal-to-noise ratio should be better discussed and underlined with corresponding additional measurements.

Author Response

Response Letter to Reviewer #2

We sincerely thank you for giving us such valuable suggestions for revision. The reviewers′ comments are indeed constructive and helpful for us to improve our manuscript. Therefore, we correctly revised the manuscript based on the reviewers′ comments. The following is our point-to-point response to the reviewers′ concerns and descriptions of the revisions, highlighted in yellow in the revised manuscript:

[Comment 1]

The authors introduced a few improvements to the manuscript. They present an interesting approach for sodium ion sensing and propose applications in the field of biosensing. The dual gate configuration has already been published. The novelty of the article is the introduction of the sodium selective membrane which is sparsely discussed. Especially a physiological context is completely missing. What are physiological relevant concentrations, is the sensor stable enough for the detection of physiological relevant changes, LOD, noise and the signal-to-noise ratio should be better discussed and underlined with corresponding additional measurements.

[Answer 1]

We admired the reviewer's sharp and helpful questions. Physiologically relevant concentrations, detection limits, noise, and signal-to-noise ratios mentioned by reviewers are important factors in sensor platforms. Therefore, we additionally measured 0.01 mM and 200 mM NaCl to evaluate detection limits and physiologically relevant concentrations, respectively. As a result, it was confirmed that the detection limit of the sodium-ion sensor proposed in this paper was about 10-4 M. In addition, since the fabricated sodium ion sensor showed excellent linearity and high sensitivity including 100 mM and 200 mM, it is expected to be able to analyze physiological stress in blood or sweat. In addition, the content related to noise and signal-to-noise mentioned in previous response letter has also been added to the manuscript.

In response to the reviewers' comments, we have also revised the manuscript in the following section for the reader's understanding:

(Line 189 – Line 196, page 6 in revised manuscript)

Additionally, to investigate the effect of noise, we tested the stability of both modes, as shown in Figure 6, by applying a 0.4 V amplitude square wave pulse (assuming noise) to the gate voltage sweep. As a result, Imax/Imin = 78.36 in SG mode, while Imax/Imin = 1.3 in DG mode. This is because the SiNW channel has an SS of 713.8 mV / dec in the transfer characteristic curve in the DG mode, which is larger than that of 96.9 mV / dec in the SG mode, resulting in a relatively small change in the drain current with respect to the pulse (noise) of the gate voltage. Therefore, we demonstrated that SiNW channel DG FETs have better noise tolerance than conventional SG FETs.

(Line 258 – Line 261, page 9 in revised manuscript)

As can be seen in Figure 8(a) and (e), the transfer characteristic curves shifted with a constant interval only in the rage of 10-4 − 100 M. However, it showed poor linearity at 10-5 M, meaning that the detection limit is about 10-4 M. Therefore, we extracted the sensitivity in the range of 10-4 − 100 M.

(Line 266 – Line 274, page 9 in revised manuscript)

Extracting by inversely calculating the Na detection capacity for mV yields SG = 1.74 × 10-2 dec/mV and DG = 6.85 × 10-4 dec/mV. The goal of sensitivity amplification is to improve the corresponding resolution, i.e., to reduce the minimum change in concentration of sensitivity. Therefore, in terms of the sensitivity resolution of the sodium ion sensor, it is obvious that the DG mode operation based on capacitive coupling is much larger than the SG mode operation. Furthermore, the linearity of the fitting line, meaning sensitivity, was 99.7% and 99.8% in SG mode and DG mode, respectively. Therefore, the sodium ion sensor proposed in this study showed excellent linearity and high sensitivity with respect to physiological concentrations including 100 mM and 200 mM.

(Line 282 – Line 289, page 9 in revised manuscript)

In addition, based on a comprehensive theoretical analysis of the DG structure, previous studies have reported that DG FET-based sensors outperform SG FET-based sensors in terms of signal-to-noise ratio (SNR) and resolution [22]. The SNR of a sensor is expressed as a ratio of its sensitivity to noise, where the DG FET outperforms the SG FET in terms of SNR when the noise of the instrument is greater than the intrinsic noise of the SG FET or DG FET. This is because the DG FET features higher sensitivity even though the instrument noise is the same for both devices. Therefore, the SiNW channel DG FET achieves high sensitivity, serving as a stable sensor platform against noise.

Again, thank you for your kind consideration and significant advice of our manuscript.

Sincerely,

Won­-Ju Cho, Professor

Department of Electronic Materials Engineering

Kwangwoon University

20, Gwangun-ro, Nowon-gu, Seoul, 01897, Republic of Korea
